# Consideration of stiffness of wall layers is decisive for patient-specific analysis of carotid artery with atheroma

Ondřej Lisický[1]*, Aneta Malá[2], Zdeněk Bednařík[3], Tomáš Novotný[4], Jiří Burša[1]

**1** Institute of Solid Mechanics, Mechatronics and Biomechanics, Brno University of Technology, Brno, Czech Republic, **2** Institute of Scientific Instruments, The Czech Academy of Science, Brno, Czech Republic, **3** 1st Department of Pathology, St. Anne's University Hospital Brno and Faculty of Medicine, Masaryk University, Brno, Czech Republic, **4** 2nd Department of Surgery, St. Anne's University Hospital Brno and Faculty of Medicine, Masaryk University, Brno, Czech Republic

ↄ These authors contributed equally to this work.
* 161238@vutbr.cz

**Data Availability Statement:** All supporting data (models, statisticas) are available on DOI: https://doi.org/10.6084/m9.figshare.12937196.

## Abstract

The paper deals with the impact of chosen geometric and material factors on maximal stresses in carotid atherosclerotic plaque calculated using patient-specific finite element models. These stresses are believed to be decisive for the plaque vulnerability but all applied models suffer from inaccuracy of input data, especially when obtained *in vivo* only. One hundred computational models based on *ex vivo* MRI are used to investigate the impact of wall thickness, MRI slice thickness, lipid core and fibrous tissue stiffness, and media anisotropy on the calculated peak plaque and peak cap stresses. The investigated factors are taken as continuous in the range based on published experimental results, only the impact of anisotropy is evaluated by comparison with a corresponding isotropic model. Design of Experiment concept is applied to assess the statistical significance of these investigated factors representing uncertainties in the input data of the model. The results show that consideration of realistic properties of arterial wall in the model is decisive for the stress evaluation; assignment of properties of fibrous tissue even to media and adventitia layers as done in some studies may induce up to eightfold overestimation of peak stress. The impact of MRI slice thickness may play a key role when local thin fibrous cap is present. Anisotropy of media layer is insignificant, and the stiffness of fibrous tissue and lipid core may become significant in some combinations.

## Introduction

Atherosclerosis is a cardiovascular disease causing local intimal thickening of artery wall and plaque formation. Vulnerable atherosclerotic plaques, characterized by lipid accumulation under a thin fibrous cap (FC), have attracted attention of researchers for more than two decades [1, 2]. A rupture of the plaque may cause blood clot formation and leads to the stroke in case of carotid arteries [3]. The rupture occurs when stresses induced by mechanical loading

**Funding:** This work was supported by Czech Science Foundation (https://gacr.cz/en/) project No. 18-13663S. The MRI service was funded by project LM2015062 "National Infrastructure for Biological and Medical Imaging (Czech-BioImaging)" of the Ministry of Education, Youth and Sports of the Czech Republic (http://www.msmt.cz/?lang=2).

**Competing interests:** No authors have competing interests.

**Abbreviations:** 2D, two-dimensional; 3D, three-dimensional; CCD, central composite design; CT, computed tomography; DoE, design of experiment; FC, fibrous cap; FE, finite element; FSI, fluid structure interaction; FT, fibrous tissue; LC, lipid core; MB, media behaviour; MRI, magnetic resonance imaging; PCS, peak cap stress; PPS, peak plaque stress; PS, patient specific; SEDF, strain-energy density function; ST, slice thickness; WT, wall thickness.

exceed the tissue strength. Thus, the peak cap stress (PCS) within the FC is believed to be the most valuable indicator of the plaque vulnerability [4–7].

Recent progress in imaging techniques and in computational modelling enables us to use patient-specific (PS) geometries for assessment of stresses. Finite element (FE) analysis [6, 8] or fluid-structure interaction (FSI) [9–12] are mostly used to analyse the impact of key factors like geometrical parameters or mechanical characteristics of the plaque and arterial wall on the stresses.

Like any diseased arterial wall, the thickened intima layer called fibrous tissue (FT) is very heterogeneous. Hence its mechanical properties vary locally within a plaque as well as among different plaques [7, 13, 14]. Therefore, analyses of plaque composition and experimental investigation of mechanical properties of its components are needed. In particular, little information is known about mechanical behaviour of lipid core (LC). Frequently its very soft behaviour is assumed in computational modelling [4, 6, 15], except for a study applying the other extreme of very high stiffness [16]. In contrast, mechanical behaviour of the vessel wall layers appears to be well supported by experiments. Uniaxial and biaxial mechanical tests of separated wall layers [8, 17, 18] enable us to describe its anisotropic behaviour even accounting for the wall structure.

Simplified 2D models used earlier in FE analyses were found to overestimate the stresses [6, 10, 19], therefore the research should focus on 3D models [20]. However, the level of these models varies. Simpler models with idealized geometry use to consider either homogeneous or layered arterial wall [4, 21], although the omission of the wall is also presented frequently [7, 10, 12]. Nowadays PS models are preferred but they suffer from lack of geometrical data and large dispersion of mechanical properties. While 2D histology sections or *ex vivo* MRI scans of autopsy samples enable us to reconstruct completely both the lumen and the vessel wall [4, 6, 8], the MRI images taken either *in vivo* [9, 12] or with samples from endarterectomy (where the plaque is resected without media and adventitia) do not enable us to capture wall boundaries. Consequently, the 3D models based on *in vivo* imaging give often contradictory results depending, for instance, on consideration of real mechanical properties of outer wall layers or FT stiffness [4, 7] or quality of the geometrical data [6, 22].

In this study, 3D PS FE models of carotid atherosclerotic artery for two patients accounting for both media and adventitia layers are used to investigate the significance of several factors in the calculation of extreme stresses in the atheroma; this should indicate their significance in further vulnerability diagnosis and enable us to propose a reasonable level of computational models. In total, five different factors related to mechanical properties or geometry of the atheroma are investigated; application of the design of experiment (DoE) concept, especially central composite design (CCD), resulted in one hundred solved computational models.

## Methods

### Acquiring data on geometry and structure

**Ethics statement.** This study was approved by the medical ethical committee of St. Anne's University Hospital in Brno (reference number 12V/2017). Written informed consent was obtained from the subject.

**Magnetic resonance imaging (MRI).** Atherosclerotic plaques were harvested (sample 1: man, 61 years, sample 2: man 71 years) during endarterectomy in St. Anne's University Hospital in Brno which met the following requirements: (1) the presence of at least one large LC with calcifications within, (2) undamaged and entire external surface of the FT and (3) an overall length which could be covered with a sufficient number of images using standard *in vivo* axial resolution. The samples were classified as an atherosclerotic lesion of Type VII [23].

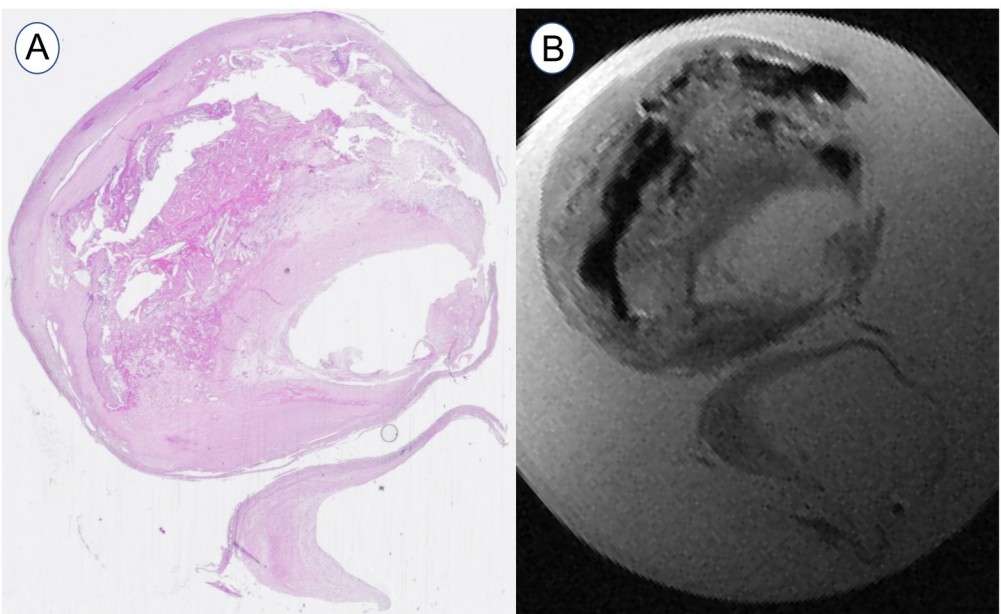

**Fig 1. Example of carotid plaque bifurcation.** From the histology cross-section (A) and the high-resolution MRI 9.4 T with ST = 0.25 mm (B).

Post-operative imaging of the samples was performed with a 9.4 T MRI system (Bruker BioSpec 94/30 USR, Ettlingen, Germany). The images were acquired in two series: the first one with the RARE technique (repetition time TR = 2500 ms, effective echo time TEeff = 10.77 ms), the second one with IR-RARE (TR = 3000 ms, TEeff = 6.66 ms, inversion time TI = 950 ms). The slice thickness (ST) was set to 1.5 mm, 1 mm and 0.25 mm for the purpose of comparison. An in-plane resolution of 78 μm was constant for all measurements with the matrix size of 256x256 (see Fig 1).

**Histology.**   After MRI the samples were fixed in 10% neutral formalin for 24 hours and then subjected to decalcification, dehydrated and embedded in paraffin. Subsequently, 3–5 μm thick tissue slices were cut off and stained with hematoxylin-eosin. Then the slices were scanned and compared with the individual MRI images to distinguish the tissue components properly (see Fig 1). However, they could not be directly used for model segmentation due to their unavoidable deformation after the histological processing [22].

## 3D PS model of carotid artery with atheroma

Semi-automatic segmentation of the recorded MR images was applied using RETOMO (BETA CAE Systems) to distinguish the plaque components. We focused on segmentation of the FT and LC (including the regions of calcification). As biological structures are characterized by smooth surfaces [8], the geometry was smoothed [24] with the same parameters for all the models. Since the individual vessel wall layers cannot be distinguished in the *in vivo* MRI scans and the sample harvested at endarterectomy cannot include the media and adventitia layers of the wall, the outer surface of the obtained geometry was used to create the volume of the missing layers. For this purpose, the outer surface of the specimen was offset equidistantly by 0.5 mm [11, 21] to create the outer surface of the artery wall. Although it is known the media may be degraded and non-uniformly thick under the plaque, the assumption of constant wall thickness (WT) was adopted here as an appropriate simplification and the thickness

was included into DoE as one of the investigated parameters. The reconstructed model is depicted in Fig 2.

## Material models

Media was modelled as a fibre-reinforced anisotropic layer, composed of a non-collagenous matrix and two families of collagen fibres. Based on mean material response [25], its anisotropic behaviour was characterized by the following strain-energy density function (SEDF) according to [26]:

$$\Psi = \Psi_{iso} + \Psi_{aniso} = \frac{\mu}{2} + \frac{k_1}{2k_2}\left(e^{k_2(1-\rho)(I_1-3)^2+\rho(I_4-1)^2} - 1\right)$$

(1)

where $\mu > 0$ is a stress-like parameter describing the isotropic response of the tissue, $k_1 > 0$ is a stress-like parameter related to the collagen fibre stiffness, the dimensionless parameter $k_2 > 0$ refers to the level of fibre strain stiffening, $I_1 = \lambda_r^2 + \lambda_\theta^2 + \lambda_z^2$ and $I_4 = \lambda_\theta^2\cos^2\varphi + \lambda_z^2\sin^2\varphi$ are invariants of the right Cauchy-Green deformation tensor $\mathbf{C}$ ($I_4$ related to stretches of two fibre families), and $\rho < 0;1>$ is a concentration parameter representing a "degree of anisotropy". In the applied model, + or -$\varphi$ denotes the angle between the circumferential direction and direction of each fibre family located symmetrically in the tangential plane of the tube. As the applied FE software Ansys does not offer this material option, the model was implemented via a user material subroutine.

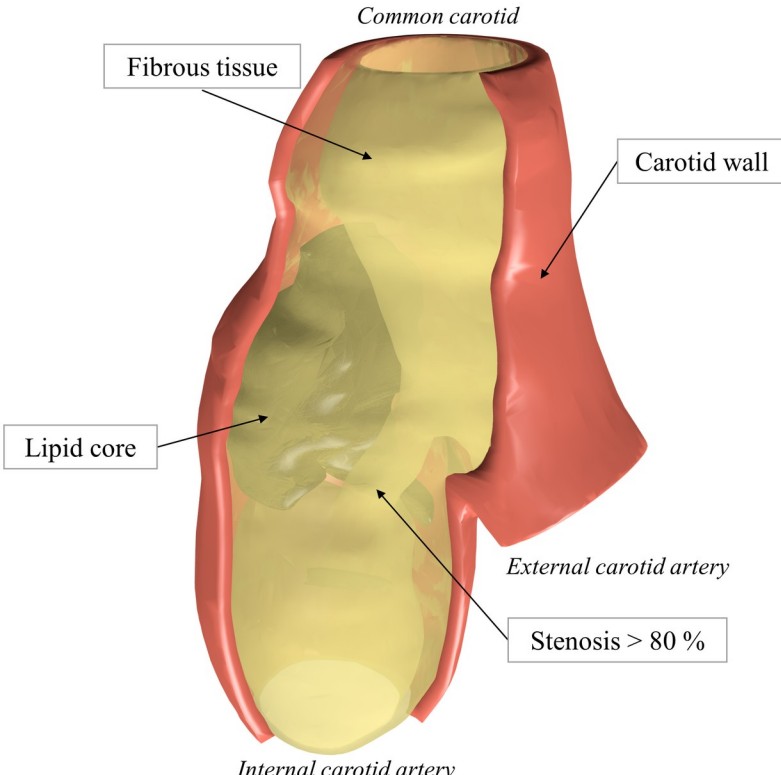

*Common carotid*

Fibrous tissue

Carotid wall

Lipid core

*External carotid artery*

Stenosis > 80 %

*Internal carotid artery*

**Fig 2. Illustration of the 3D PS model of the atherosclerotic carotid plaque (sample 1) obtained from endarterectomy.** Carotid wall (red) shares the surface with FT.

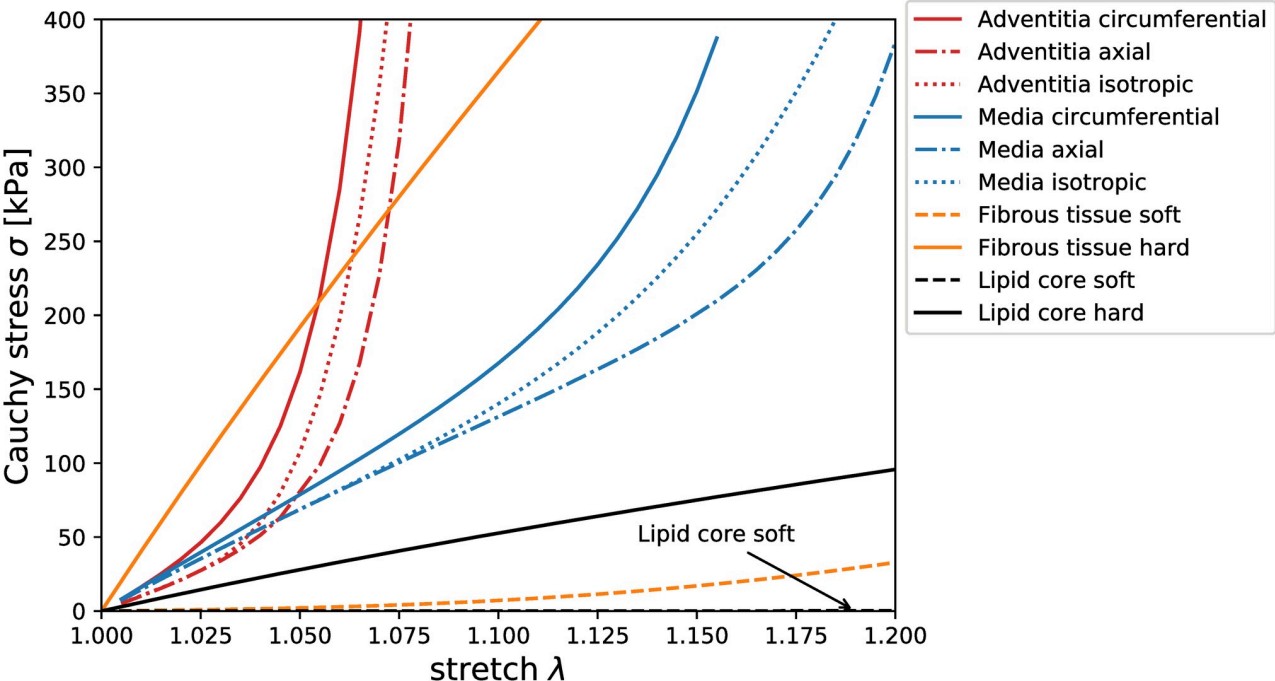

**Fig 3. Stress-stretch equi-biaxial responses of the chosen constitutive models.** The dotted lines show the isotropic fit of the circumferential (solid) and axial (dash dotted) anisotropic responses. The soft (dashed) and the hard (solid) isotropic responses for the FT and LC are also presented for comparison.

The other wall layers and plaque components were modelled as incompressible, hyperelastic and isotropic since their anisotropy is low [17] compared to the inter-patient variability; this simplification is generally accepted [27]. To fit the biaxial response of the adventitia (and also of the media when modelled as isotropic for comparison), 3rd order Yeoh type SEDF [28] was used in the following form:

$$\Psi = \sum_{i=0}^{3} c_{i0}(I_1 - 3)^i \qquad (2)$$

where $c_{i0}$ are material parameters. Material responses of both layers were taken from experimental biaxial testing of each carotid component [17]. Material parameters for the wall layers were identified with considering residual stresses and stretches observed in the load-free geometry [25]. Uniaxial test data of FT [7, 13, 14] were fitted with the 2nd order Yeoh SEDF while the LC was modelled with Neo-Hookean (i.e. 1st order Yeoh) constitutive model. Material responses can be seen in Fig 3 with the used material parameters presented in Table 1.

## FE model setup and boundary conditions

FE analysis was performed in FE software ANSYS 19.2. (Ansys Inc., PA, USA). The vessel wall was meshed with four/eight elements across the WT using linear 8 node hexahedral elements (SOLID185) in ICEM 19.2. (Ansys Inc. PA, USA) see Fig 5. A linear tetrahedral element (SOLID285) mesh performed in ANSA (Beta CAE) was used for the FT and LC. A bonded contact was used to join the FT with the vessel wall, while the other components were connected by shared nodes. A constant pressure value of 13 kPa (mean arterial pressure) was applied on the luminal surface of the geometry. Free ends of the vessel wall were fully

**Table 1. Material parameters.**

| Tissue | Type | Material constants [a] | | | | |
|---|---|---|---|---|---|---|
| Media | A | $\mu = 122.3$ | $k_1 = 24.7$ | $k_2 = 16.5$ [-] | $\varphi = 6.9$ [°] | $\rho = 0.8$ [-] |
| | I | $c_{10} = 122.3$ | $c_{20} = 0$ | $c_{30} = 337.7$ | | |
| Adventitia | I | $c_{10} = 88.7$ | $c_{20} = 0$ | $c_{30} = 45301.4$ | | |
| Lipid core | I soft | $c_{10} = 0.1$ | | | | |
| | I hard | $c_{10} = 50$ | | | | |
| Fibrous tissue | I soft | $c_{10} = 2.7$ | $c_{20} = 20$ | | | |
| | I hard | $c_{10} = 342.1$ | $c_{20} = 20$ | | | |

[a]The above values are in kPa if not otherwise specified. Symbols I and A denote the isotropic and anisotropic behaviour, respectively.

constrained. A mesh convergence analysis was performed to obtain a suitable element size for all models. A typical mesh density resulted in 24k elements for the vessel wall and 95 k for FT with LC.

**Material orientation.** Consideration of media layer anisotropy drew the problem of the orientation of principal material axes in each point of the model. Each element coordinate system needs to be rotated to the preferred orientation of the collagen fibres. In this study, we take advantage of the applied locally structured hexahedral mesh which enables us to define a centroid of each section and subsequently also the central line (see Fig 4). Two points of this

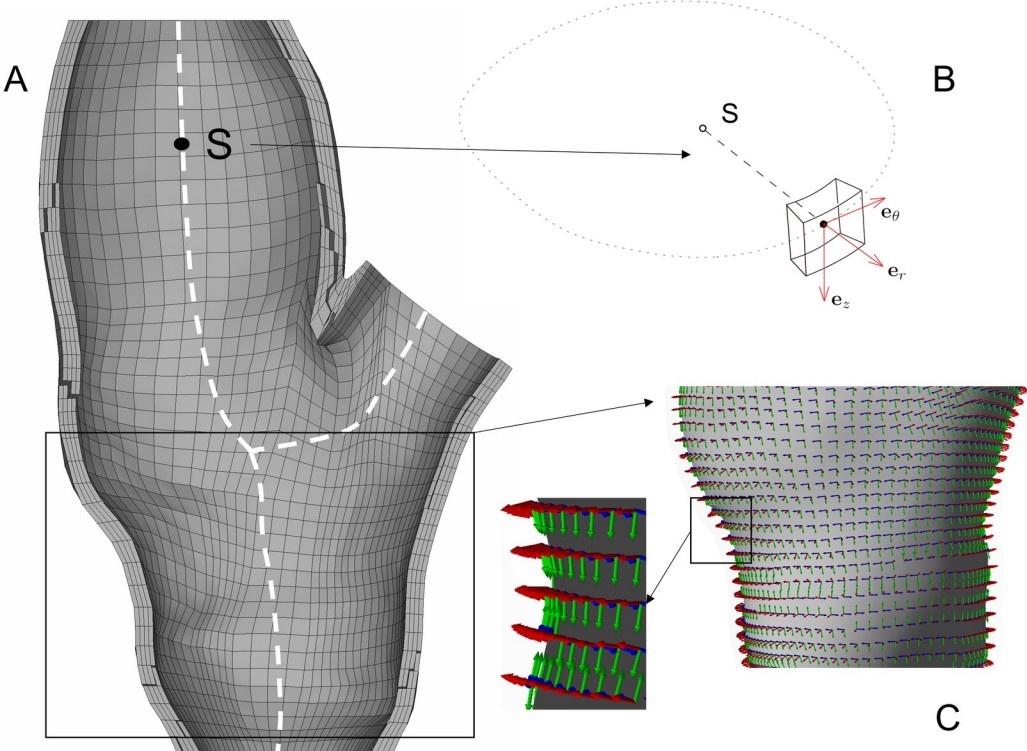

**Fig 4. Computational model with principal material directions.** (A) The discretized model of the vessel wall of sample 1 with hexahedral elements; the central line (white dashed) serves for definition of the axial base $e_z$. (B) Detailed illustration of the coordinate system of an individual element. (C) Detail of element-specific material directions within the vessel wall model; the blue, green and red arrows indicate the circumferential, axial and radial directions, respectively.

line close to the section were used to define the local axial basis $\mathbf{e}_z$. A vector normal to the inner surface of the element located in its centroid was used as the local radial basis $\mathbf{e}_r$. After that, a cross product of those two bases resulted in the circumferential basis $\mathbf{e}_\theta$. The resulting distribution of bases determinates the principal material directions when the anisotropic constitutive model is applied.

## The investigated factors and their statistical analysis

In biomechanics, all the input data for computational modelling suffer from large uncertainties or inaccuracies. This variance of input data may influence the character of the deformation and consequently, the stresses evaluated as a potential indicator of plaque vulnerability. Similarly to another biomechanical study [29], DoE with 95% confidence level was applied for this statistical analysis; specifically the CCD was chosen, to analyse (using Minitab 15 statistical software) the possible non-linear dependence of PCS on some of the five selected factors. This resulted in overall 100 computational models for the two patients, differing in their material and geometrical factors; these models afforded the input data for statistical analysis.

In the DoE methodology, a 2-factorial design was performed and thereafter augmented with a set of axial points resulting in the CCD, specifically the face-centred design. Here, the middle level of each factor is used (centre of a factorial face) enabling to estimate a possible curvature of the response. The model suitability was checked for all the fitted responses via normal distribution of residuals and a sufficient coefficient of determination ($R^2$>0.8). The first (maximal) principal stresses were adopted as stress response indicators, especially PPS and PCS [4, 6]; for all the 3D models analysed, they were evaluated in cross-sections with a 0.2 mm span. Lastly, the analysis of variance was performed within the DoE to find the significance of the individual factors and of their combinations. Except for media behaviour (MB), all the factors were set as continuous. A specific choice of factors can be found below with their range of values summarized in Table 2.

**Lipid core stiffness.** The LC plays a key role in the plaque vulnerability through its size or location in the plaque where the resulting distance from the lumen to the LC is characterized as FC thickness [7, 21]. However, the LC composition almost disables its mechanical testing. Thus, the stiffness of the LC was chosen as a variable factor ranging from 0.1 to 50 kPa. The lower value (related to lipids) is mostly used in computational models while the upper value was chosen to reflect the hypothesis that micro-calcifications occurring in the extracellular matrix of the LC may cause its stiffening by orders.

**Fibrous tissue stiffness.** Experimental data of carotid plaques [7, 13, 14] were analysed to find samples with extreme stiffness which were then fitted with the second-order Yeoh SEDF (see Fig 3). Nearly linear stress-strain response, i.e. a very low strain stiffening, was found for both the most and the least stiff plaques, thus the number of variables could be reduced. The second material parameter $c_{20}$ (related to strain stiffening) was fixed and only the first constant $c_{10}$ (initial stiffness) was fitted giving a sufficient quality of the fit ($R^2$>0.8).

**Table 2. DoE factors.** Selected factors for DoE with their chosen upper and lower limits.

| Factor | Minimum | Maximum |
|---|---|---|
| Wall thickness [mm] | 0 | 0.5 |
| Slice thickness [mm] | 0.25 | 1.5 |
| Lipid core $\mu$ [kPa] | 0.1 | 50 |
| Fibrous tissue $c_{10}$ [kPa] | 2.7 | 342.1 |
| Media behaviour | isotropic | anisotropic |

**Media behaviour.** The biaxial response of carotid media [25] was used to fit both anisotropic and isotropic constitutive models (see Fig 3). The anisotropic description was applied with the locally varying orientation of the element coordinate system.

**Slice thickness.** *In vivo* and *ex vivo* imaging methods differ substantially in their resolution (both axial and in-plane). Histological slices are often used for reconstruction of 2D and 3D PS plaque geometries [4, 6] since they provide a suitable in-plane resolution. However, it is very laborious to achieve a sufficient axial resolution and the deformation caused by slice preparation [20, 22] may result in distortion of the shape. Therefore, *ex vivo* MRI was used for the 3D reconstruction in this study. As only *in vivo* imaging has potential in clinical assessment of plaque vulnerability, the impact of different resolutions of the MR images was chosen as another investigated factor, since the lower axial resolution of the *in vivo* MRI may cause loss of geometrical information significant for the computational analysis [6]. This factor was set as continuous with values ranging from 0.25 (resolution of the applied *ex vivo* MRI) to 1.5 mm (a typical resolution of *in vivo* MRI).

**Vessel wall thickness.** The vessel WT was set from 0 to 0.5 mm [11, 21] where the lower limit represents models with identical mechanical properties for the wall and the FT [7, 12, 19], neglecting thus the impact of different properties of the outer arterial layers. This means the volume of the wall was always included in the model but with different material properties. In all the models the thickness of the wall was halved between media and adventitia. Fig 5 shows the used configurations where the vessel wall is represented with 4 elements per thickness even when the half thickness t = 0.25 mm (axial point) is considered (see Fig 5).

Since the CCD requires only continuous variables, the discrete MA factor was fixed and the DoE was performed twice with four continuous factors. The DoE results were compared using a paired t-test to investigate the significance of the MB factor.

## Results

### The factors significant for the statistical model are summarized in the Table 3.

An example of the numerical results of the PS model of the atherosclerotic carotid artery is presented in the form of displacements and first principal stresses for sample 1 in Fig 6. Cross-

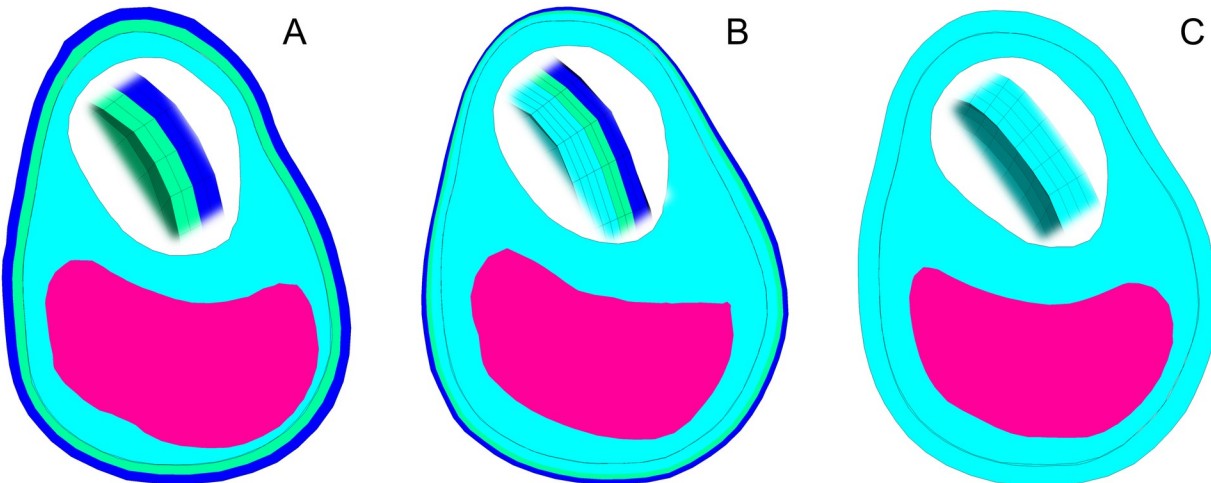

**Fig 5. Cross-sections representing the model configurations.** Different WT of 0.5 mm (A), 0.25 mm (B) and 0 mm (C). The adventitia (dark blue) is always represented with 2 elements, as well as the media (light green). FT (light blue) increases its volume with decreasing WT. LC representation (pink) is immutable in the same axial position and with the same ST. Details of the wall are presented in the white lumen area.

**Table 3. Analysis of variance for the PPS and PCS.**

| Sample 1 | Peak Plaque Stress | | | | | | | | |
|---|---|---|---|---|---|---|---|---|---|
| | | Linear | | Square | 2-Way interaction | Model R² | | | |
| | Factor | **WT** | FT | **WT*WT** | **WT*FT** | 0.894 | | | |
| | p-value | **<0.001** | 0.48 | **0.08** | **<0.001** | | | | |
| | Peak Cap Stress | | | | | | | | |
| | | Linear | | | Square | 2-Way interaction | | | Model R² |
| | Factor | **WT** | **LC** | FT | LC*LC | **WT*LC** | **WT*FT** | **LC*FT** | 0.845 |
| | p-value | **<0.001** | **0.01** | 0.117 | 0.124 | **0.072** | **<0.001** | **0.038** | |
| Sample 2 | Peak Plaque Stress | | | | | | | | |
| | | Linear | | | | 2-Way interaction | | Model R² | |
| | Factor | ST | WT | LC | FT | WT*FT | LC*FT | 0.874 | |
| | p-value | **0.025** | **0.001** | 0.113 | 0.115 | **<0.001** | 0.146 | | |
| | Peak Cap Stress | | | | | | | | |
| | | Linear | | | | Square | 2-Way interaction | | Model R² |
| | Factor | **ST** | **WT** | **LC** | **FT** | **LC*LC** | WT*LC | **WT*FT** | 0.823 |
| | p-value | **0.026** | **<0.001** | **<0.001** | **0.015** | **0.018** | 0.122 | **<0.001** | |

The table shows the statistical significance of factors included in the statistical model represented by their p-values and R². The statistically significant or nearly significant factors and their combinations are highlighted in bold.

sections in the selected locations show typical stress distributions focused on the PPS and PCS. Results near the boundary conditions and the bifurcation were not considered since they may cause non-realistic artefacts (stress concentrations due to boundary conditions). The discrete media-related factor MB was found to be insignificant (probability value p = 0.99) with stresses showing a mean difference of 1 kPa between the models with isotropic and anisotropic media for both patients. Therefore, this factor was not included in the evaluations below.

## Peak plaque stress

The statistical model fitted to the PPS response showed the same significance of the WT for both samples. Unlike in sample 1, ST was found significant in sample 2. Also, the interaction between FT stiffness and WT (FT*WT) was found significant in both cases. The squared effect, evaluated by the additional face-centred axial points, was significant for sample 1 in case of WT factor. The maximum was mostly localized in the narrow part of the FT component between the luminal surface and the inner surface of the vessel wall, except for the cases with lower stress concentration where the maximum was located within the FC (see detail D in Fig 6). The cube plot in Fig 7 shows the PPS responses at low and high levels of each factor. In the model with soft FT, the PPS increased on average up to five times (from 29 to 160 kPa for sample 1 and 44 to 230 kPa for sample 2) when the wall was not included. In contrast, an opposite tendency occurred when the hard FT was considered; the PPS decreased by some 30% when the wall was not included.

 The highest stress concentrations (168 kPa and 313 kPa) were found in the models without the vessel wall and with the soft atheroma components, while the lowest stresses (22 kPa and 26 kPa) were found in the models with the artery wall included and with the soft FT and hard LC. The results show that similarly to the MB factor, neither LC stiffness nor FT stiffness itself has a significant impact on the PPS minimum/maximum as long as interactions are not considered. For all the responses with the artery wall included, however, Fig 7 shows 3 up to 5

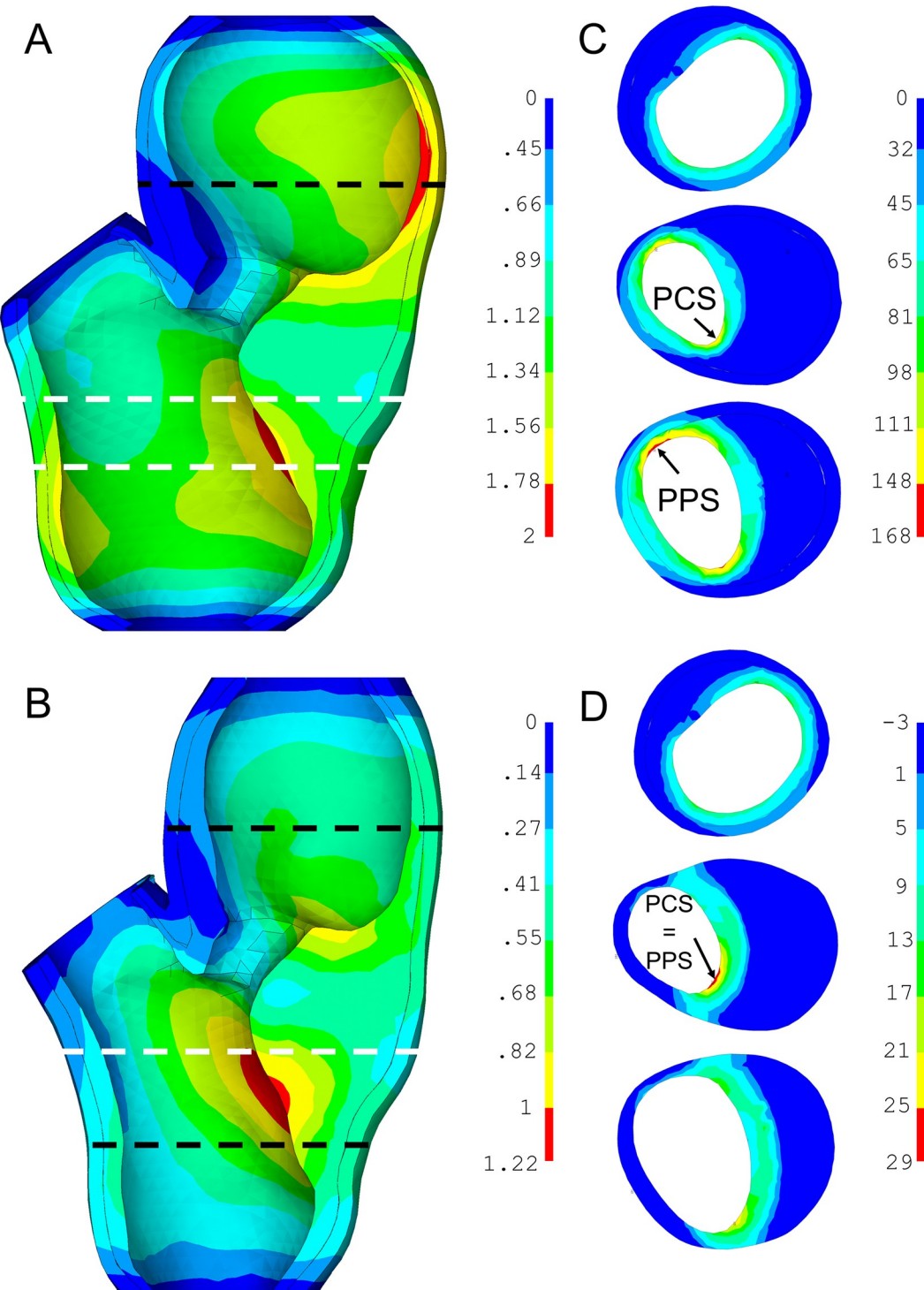

**Fig 6. Contour maps of two sample 1 models.** Both LC and FT being soft, differing in WT only: WT = 0 mm (A), (C) and WT = 0.5 mm (B), (D). Displacements in (A) and (B) show their 64% overestimation in model (A) with media and adventitia properties replaced by those of the FT. The maps of first principal stresses (C) and (D) show their distribution in the chosen locations (dashed lines) with the arrows indicating maximal PPS and PCS in the respective sections (white dashed lines). In the model with media and adventitia replaced by the FT, the maximal stresses are almost six times higher. Note that (D) is shown without the media and adventitia layers due to very different levels of stresses between the plaque and these layers.

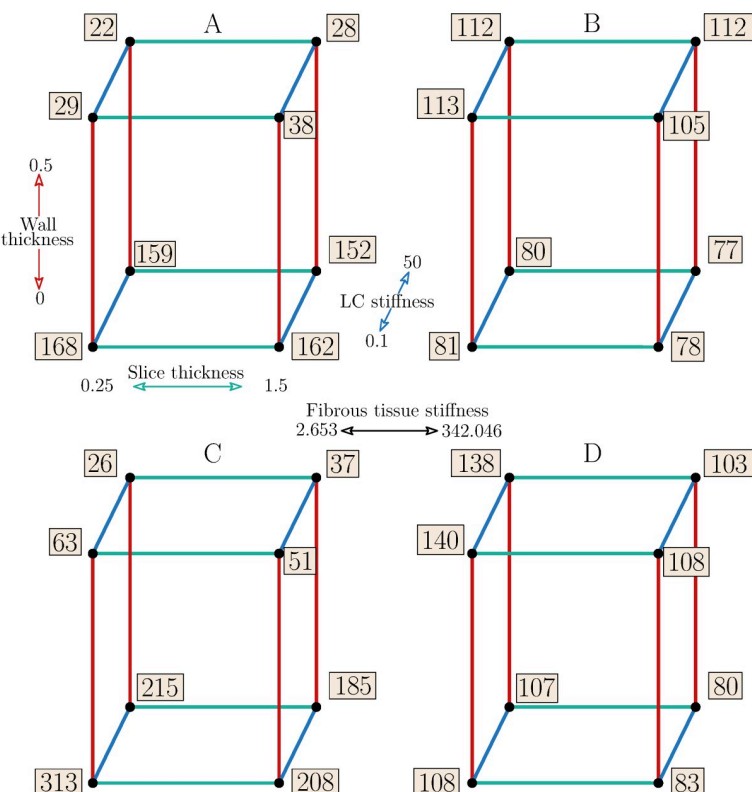

**Fig 7. Cube plots of PPS values [kPa].** Patient 1 (A-B) and patient 2 (C-D). Colours represent a specific factor; red—WT, green—ST, blue—LC stiffness. Cubes A and C represent low stiffness FT while B and D high stiffness FT.

times higher PPS values for the hard FT compared to the models with the soft FT, except for some lower ratios occurring when half WT is considered.

## Peak cap stress

Maximum PCS was localized in the shoulder region of the FC in the majority of cases (see Fig 6). In contrast to PPS, the impact of the FT stiffness was found significant for sample 2, while that of the LC stiffness plays also a significant role in some cases. Also here, however, the main effect is caused by the WT and by the interaction of WT with the FT. ST and squared effect of LC factor were found significant for sample 2 although they showed only minor effect for sample 1. The PCS values at low/high levels of all the investigated factors are summarized in Fig 8. The highest PCS differences occur for the soft FT between the cases with WT = 0 and WT = 0.5, while with the hard FT the response was nearly independent of the other factors. The highest stress concentrations (153 kPa and 300 kPa) were found in the models without the vessel wall and with the soft atheroma components, while the lowest stresses (22 kPa and 25 kPa) were found in the models with the artery wall included and with the soft FT and hard LC. The maximum difference in the PCS between soft and hard components (both FT and LC) was 29 kPa vs 47 kPa and 57 kPa vs 53 kPa with the WT fully included while it amounted to 138 kPa vs 49 kPa and 300 kPa vs 50 kPa without the wall; this comparison was based on results for the minimal slice thickness only.

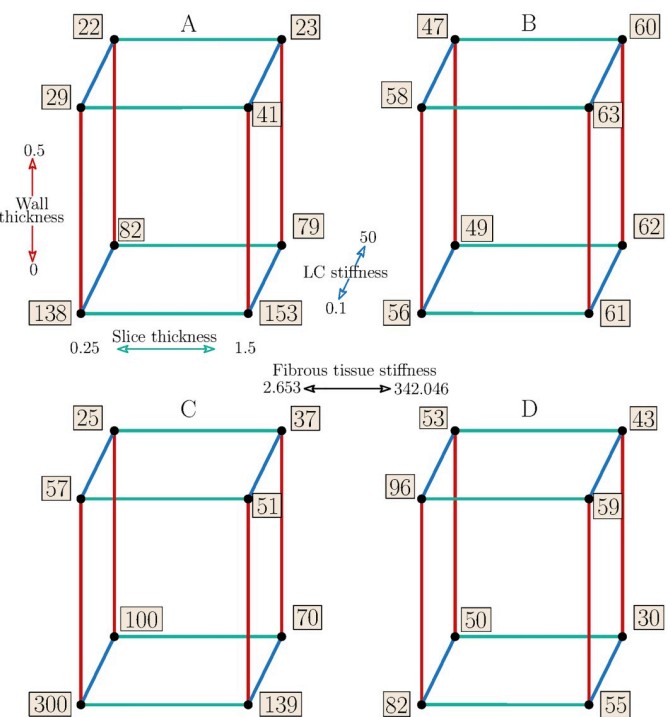

**Fig 8. Cube plots of PPS values [kPa].** Patient 1 (A-B) and patient 2 (C-D). Colours represent a specific factor; red—WT, green—ST, LC stiffness—blue. Cubes A and C represent low stiffness FT while B and D high stiffness FT.

## Discussion

Credibility of computational modelling in the assessment of atherosclerotic plaque vulnerability depends on a considerable number of factors the uncertainty of which influences the calculated stress distribution throughout the plaque; evidently, their impact is worth investigating when a realistic response is required. A great number of studies focused on geometrical parameters and indicated the FC dimensions as crucial parameters for the plaque vulnerability [4, 7, 11, 21]. Even though the presented models did not fully meet the vulnerable plaque definition, it is not clear whether such plaques are the only ones which should be investigated for high stresses by computational modelling in order to examine a mechanism of rupture. Recent study Libby *et al.* [30] evaluated the current strict focus on vulnerable plaques with the conclusion that we should be prepared to address its fundamental causes in greater depth. Burke *et al.* [31] defined the limit FC thickness for vulnerable plaques <65 μm which is below the resolution of *in vivo* or even *ex vivo* imaging methods (except for e.g. μCT or histology) and so applicable analysed only to idealized studies. Moreover, stresses found in some configurations within our study exceeded a tissue threshold of 300 kPa which was discussed e.g. in Holzapfel *et al.* [20]. This indicates that either local thresholds need to be defined or that even non-vulnerable plaques from its definition might be critical for rupture risk. In this study, we focused on consideration of material properties of the vessel wall along with those of the atheroma; we used 3D PS computational models and the DoE strategy to investigate their influence on PCS and PPS.

### Wall thickness

To the authors´ best knowledge, there is no current study investigating the possible impact of the wall consideration in the computational model of the atherosclerotic plaque on the stress

distribution even though the omission of the wall is very frequent in *in vivo* studies; therefore, different studies will be discussed. To consider the composition of the wall with atheroma in our study, the vessel wall was always included, although with different material properties. In Huang *et al.* [10] the authors used a PS plaque model without distinguishing between the vessel wall components. When comparable inputs are taken into consideration, the PCS maxima correlate with our findings (120 kPa vs 58 to 147 kPa represented by average values in our study depending on the FT stiffness). Yuan *et al.* [7] presented an idealized plaque model not distinguishing between the wall components but with various stiffness for each plaque component based on its mechanical tests. The decrease in the FT stiffness resulted in an increase of PCS and PPS which correlates with our findings using the configuration with WT = 0. Maximum PCS of 228 kPa was found for the configuration with a thick FC, cf. our average value of 183 kPa with comparable model inputs and independent of ST. Consideration of the wall was presented in Nieuwstadt *et al.* [6] where lower stress concentrations (mean PCS of about 60 kPa) were found compared to Yuan *et al.* [7]. Here our findings show the average PCS values of 54 kPa with a comparable LC stiffness but different FT stiffness, the effect of which was found insignificant. When we compare the discussed studies with our findings, neglecting the specific vessel wall properties leads to higher stress concentrations which may cause incorrect evaluation of PCS. This effect is reduced with a hard FT approaching the wall stiffness itself. Especially with softer FT models, however, it holds that consideration of vessel wall is decisive for the calculated stresses while the impact of its thickness variations is much less pronounced. Specifically, zero WT resulted in the overestimation of PPS and PCS by a factor of 5 up to 8 (for the soft FT and LC models).

## Slice thickness

Study Lisický *et al.* [24] investigated the ST impact on magnitudes of surfaces and volumes of the atherosclerotic plaque components and found only slight impact of the ST in the whole investigated range between 0.25 to1.5 mm, with a negligible impact between 0.5 and 1 mm. In this work, we found that ST was significant for sample 2 while insignificant for sample 1. This bias might be related to the fact that, in case of sample 1 with the high ST, a minimum FC thickness was comparable with a low ST but in case of sample 2, there were huge differences between them. Here, the local minimum of the FC thickness was not captured by any of the images resulting in lower stress concentrations. In study Nieuwstadt *et al.* [6] the authors used different numbers of histological slices which resulted in low and high sampling models. Their results showed a large variation among different samplings though they concluded that the magnitude of PCS differed significantly when lower sampling was used. Higher ST of MRI or low sampling for other imaging methods may cause data loss if, for example, small LC or local thin FC are presented. Our results showed that the impact of ST should not be neglected and further investigation is needed. In conclusion, the *in vivo* MRI with its achievable axial resolution of at least 1.5 mm should be capable to distinguish critical plaque components, especially if equipped with a special carotid coil providing also a sufficient in-plane resolution as shown in studies [32, 33] although its resolution is still not capable to capture thin FC.

## FT stiffness

In many experimental studies, various mechanical properties of the FT were found [7, 13, 14]. The influence of FT stiffness was examined in Akyildiz *et al.* [4] where the authors used an idealized parametric model with a Neo-Hookean (i.e. nearly linear) material model for the intima and an anisotropic material model for both the media and adventitia. The soft intima models showed a lower PCS than the stiff and intermediate intima models. In our study, the same

finding was found with the increase by a factor of 1.6 to 2 between the soft and hard FT when approximately the same inputs with full WT were used. An opposite tendency (an increase of PCS with the softer FT) was found in the Yuan *et al.* [7] but this was caused by neglecting the wall components as discussed above. In our simulations with the media and adventitia replaced by the FT, the PCS also increased 2.5 to 3.6 times between the hard and the soft FT model. The explanation of this effect may be in the stiffness of the soft FT: if it is lower by orders than the stiffness of the media, adventitia or hard FT, large deformations occurring with soft FT may induce high increase of stresses. The low and high stiffness limits of the FT used in our models show a huge variance of FT; the initial stiffness of the hard FT is ca. three times higher than those of media and adventitia while this value is lower by two orders for the soft FT. Due to the complexity of the FT structure, we can expect that the variance of its stiffness may not be random but related to some other factors characterizing different experimental groups. The two or threefold increase in the extreme stresses between the low and high FT stiffness models indicates that if we were able to classify the patients (*in vivo*, for example based on MRI) into some basic groups, the variance within each group could be significantly reduced. The presence of calcifications or even their distribution [5, 34] might serve as such an indicator. Maldonado *et al.* [34], however, presented a very low (~2%) content of micro-calcifications in the FT thus we believe their impact is within the considered range of the FT stiffness.

## LC stiffness

The influence of micro-calcifications in the LC was previously discussed in [5, 34] where the authors found their high influence on the stress concentrations. Their findings indicate that thousands of micro-calcifications occur at the borders of the necrotic core or inside the LC, surrounding there some larger calcifications. Besides, the high-resolution images used in those studies indicate that modelling of calcifications as one homogeneous part [12, 16, 35] of the atherosclerotic plaque may be incorrect. Using high-resolution micro CT and histology they found that micro-calcifications are non-uniformly distributed within the FC and their amount decreases from the LC to the lumen. Grouping of these calcifications may cause artefacts in MRI and subsequent misleading evaluation of the images. In Iannaccone *et al.* [16], the whole LC was modelled in three ways: as a lipid, a fibrotic media and as a calcification. Their findings indicate that higher stresses occur when the LC is modelled as a soft lipid and correlate thus with our findings for the PCS. However, we cannot compare the results for hard LC since the LC stiffness in our models was much lower than that used in [16] where the whole LC was a ceramic-like material.

## Media behaviour

Anisotropy of the arterial wall is an undisputed fact which has been investigated in the last decades and, subsequently, numerous structure-motivated constitutive models have been formulated [26, 36, 37]. However, to take advantage of these models, information on the fibrous structure of the tissue is needed. Due to lack of information on a number of collagen fibre families and on their preferred orientation and dispersion in specific arterial tissues, these models are often fitted to mechanical experiments only which make them purely phenomenological. Also in this study, the structural parameter φ was used as a phenomenological variable [25]; there it was found to be the smallest for the media layer (<10 degrees) which indicates fibres oriented dominantly in the circumferential direction. Together with the concentration parameter ρ = 0.8, the anisotropic behaviour of the media layer can be considered as sufficiently representative even if, due to lack of information on the anisotropy of the diseased media, we used

experimental results [8, 17] obtained partly with non-atherosclerotic tissue. The results obtained in this study indicate that the isotropic representation of the biaxial tests can be used for computational modelling of the media without significant impact on stresses in the plaque or in the FC. Nevertheless, these findings do not contradict the dominating opinion that local structural parameters of collagen fibre distribution in a specific artery and arterial layer are important for the calculation of stresses in this layer and should be investigated in detail [38].

## Limitations

The FT material model may be considered as one of the main limitations of the presented study. However, no relevant data for the FT anisotropy is available, neither on its mechanical response nor on its structure. The FT extreme experimental responses selected for this study show only low strain stiffening which was captured by the second parameter of the used Yeoh SEDF. Concerning the anisotropic model of the media, the procedure applied to set its principal material directions is rigorous; their setting may be inaccurate only in very irregular parts of the geometry, such as the top of the bifurcation, where no data is available on local collagen structure. However, due to the confirmed insignificance of the media anisotropy, inaccuracies in principal directions are irrelevant. The constant pressure applied on the luminal surface might be also considered as a limitation but it was confirmed that there are no significant differences between structural and one or two-way FSI analyses [10]. Also the sample removed from the body may reduce its length and residual stresses in the tissue may be released. However, the axial pre-stretch decreases significantly with the age and thus it becomes less significant in the age group for which the atheroma is typical. Also the geometric complexity of the carotid bifurcation makes correct implementation of axial pre-stretch rather difficult, therefore it was not applied in our analyses. On the contrary, the residual stresses are believed to be a very important factor which influences the stress-strain distribution in arterial wall. In this study, the residual stresses were included in the material model of the vessel wall [25] while they were neglected in the FT and LC where, to our best knowledge, no data on their presence, magnitude, or distribution is available.

Although the geometry of the ex vivo sample may be changed, higher resolution of the MRI is substantial while distortions of the sample are negligible by virtue of its relatively high stiffness. The last limitation consists in the only two PS model geometries reconstructed in this study. Although our analyses achieved comparable statistical significance of all results (except for the MRI slice thickness which is strongly related to the complexity of geometry and its inter-patient variability), for other geometries only similar qualitative tendencies may be expected. However, a general quantitative prediction of impact of the individual factors would not be feasible, even if we made much more patient-specific calculations. Each patient has to be evaluated individually, therefore we intend to apply the presented methodology in future studies to strengthen the validity of our conclusions.

## Conclusion

Our findings point out the necessity of consideration of the vessel wall stiffness in computational modelling of atherosclerotic arteries, especially when the fibrous tissue of the plaque is less stiff than the media and adventitia layers. It appears that in FE modelling of the stress response in the fibrous cap, specifically of the peak cap stress which is believed to be decisive for plaque vulnerability, the following parameters play a key role: (i) stiffness of the fibrous tissue, (ii) stiffness of the lipid core, (iii) real material properties of the media and adventitia layers of arterial wall, especially their strain stiffening which is much more pronounced than at the fibrous tissue. In contrast, anisotropy of the media layers, as well as the wall thickness itself

(if being non-zero) are less significant for maximum stresses in the fibrous cap; this fact appears encouraging for further studies based on *in vivo* MRI.

## Supporting information

**S1 Appendix. Verification of Ansys user subroutine for media material model.**
(DOCX)

**S1 Fig.**
(TIF)

**S2 Fig. Geometry model of patient 2.** Section of sample 2 model with two large lipid cores.
(TIF)

**S1 File. Ethical agreement.**
(PDF)

## Acknowledgments

We thank to Pavel Skácel who implemented the constitutive model needed for our study into the ANSYS software.

## Author Contributions

**Conceptualization:** Ondřej Lisický, Jiří Burša.

**Formal analysis:** Jiří Burša.

**Funding acquisition:** Jiří Burša.

**Investigation:** Ondřej Lisický.

**Methodology:** Ondřej Lisický, Aneta Malá, Zdeněk Bednařík.

**Resources:** Aneta Malá, Zdeněk Bednařík, Tomáš Novotný.

**Software:** Ondřej Lisický.

**Supervision:** Jiří Burša.

**Validation:** Jiří Burša.

**Visualization:** Ondřej Lisický.

**Writing – original draft:** Ondřej Lisický.

**Writing – review & editing:** Ondřej Lisický, Aneta Malá, Zdeněk Bednařík, Tomáš Novotný, Jiří Burša.

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
