## [Decision Letter · Decision Letter 0]

11 Jun 2020

PONE-D-20-04059

Consideration of stiffness of wall layers is decisive for patient-specific analysis of carotid artery with atheroma

PLOS ONE

Dear Dr. Lisický,

Thank you for submitting your manuscript to PLOS ONE. After careful consideration, we feel that it has merit but does not fully meet PLOS ONE’s publication criteria as it currently stands. Therefore, we invite you to submit a revised version of the manuscript that addresses the points raised during the review process.

We look forward to receiving your revised manuscript.

Kind regards,

Fang-Bao Tian

Academic Editor

PLOS ONE

Journal Requirements:

Additional Editor Comments (if provided):

Reviewers' comments:

Reviewer's Responses to Questions

**Comments to the Author**

1. Is the manuscript technically sound, and do the data support the conclusions?

Reviewer #1: Yes

Reviewer #2: Partly

2. Has the statistical analysis been performed appropriately and rigorously? 

Reviewer #1: I Don't Know

Reviewer #2: No

3. Have the authors made all data underlying the findings in their manuscript fully available?

Reviewer #1: No

Reviewer #2: Yes

4. Is the manuscript presented in an intelligible fashion and written in standard English?

Reviewer #1: Yes

Reviewer #2: Yes

5. Review Comments to the Author

Reviewer #1: The authors have performed a finite element analysis of a carotid atherosclerotic plaque using an endarterectomy specimen.

They have assessed the effect of wall thickness, MR imaged slice thickness, lipid and fibrous tissue mechanical properties and media anisotropy. They have found that wall thickness and the MR slice thickness are not so critical as the mechanical properties of the lipid and fibrous tissues when computing the peak stress that might be responsible for plaque rupture.

Though it is an interesting exercise, a discussion on plaque characteristics found to be associated with plaque rupture and outcome in current literature is missing and would add critical information and perspective on the impact of their results. It is well known that a thin fibrous cap, a large lipid pool and intraplaque hemorrhage are marks of vulnerable plaque, but this markers are not included, or discussed, here.

Other comments:

Introduction

It is not clear what the author mean with the statement "While 2D histology sections or images of autopsy samples enable us ..... or with samples from endarterectomy where the plaque is resected without media and adventitia". The endarterectomy segments would be analyzed by 2D histology, or where do the 2D histology sections come from? Do the authors need to replace "or" with "of" when they say "While 2D histology sections or images"?

The introduction should include a statement indicating how their methodology is going to overcome the listed problems found in current literature.

Methods

It is unclear whether the authors scanned the endarterectomy specimen prior to histological sectioning, or before, please clarify. Also, they could comment why they did not scan the subject prior to surgery, that would have enable them to have a proper outer wall rendering.

The paragraph of the Methods section discussing the use of only one case and how the results can, or cannot, be transferred to other cases, should be moved to the limitations paragraph in the Discussion.

Discussion

The authors are encouraged to add some discussion on markers for vulnerable plaque, such as the size of the lipid pool, fibrous cap thickness and how their results would be if those were considered, or on how their methodology could accommodate for those markers.

In addition, the authors are encouraged to revise their use of English, there are some grammar and spelling errors that, at times, lead to statements that are incorrect, such as "... lead to the stroke in the carotid arteries", when stroke might be due to carotid stenosis but does not occur in those vessels, but in the brain.

Reviewer #2: Authors Lisický et al. present a manuscript titled “Consideration of stiffness of wall layers is decisive for patient-specific analysis of carotid artery with atheroma” for possible publication in the PLOS ONE. The authors have employed a Design of Experiment concept with ex vivo imaging data to investigate the effects of various parameters on plaque stress. The authors have also performed a comprehensive analysis on a number of phantom models (50 to be exact) to potentially draw a statistically meaning conclusion. In brief, the authors have highlighted that wall stiffness are more important than wall thickness in predicting plaque stress. Although the study is fundamental sound, the manuscript suffers a few shortfalls. In particular, while the authors have quantified the effects of parameters such as wall thickness, fibrous tissue stiffness, etc., the lack of patient number means that the study is, at best, hypothesis generating. It is unclear how the results would differ even with images from one additional patient. Given the highly doubt reproducibility of the present method clinically, the manuscript cannot be recommended for publication in the PLOS ONE in its current form.

Nevertheless, I have provided a list of suggestions/comments for the authors:

1. In the Abstract: “Except for anisotropy, all the other investigated factors are taken as continuous in the range based on published experimental results”. This sentence is difficult to understand. Consider rephrase.

2. On page 6: “Since the individual vessel wall layers cannot be distinguished in the in vivo MRI scans and the sample harvested at endarterectomy cannot include the media and adventitia layers of the wall, the outer surface of the obtained geometry was used to create the wall volume… equidistantly offset by 0.5 mm… the assumption of constant wall thickness (WT) was adopted here as an appropriate simplification and the thickness was included into DoE as one of the investigated parameters”. While the authors have provided suitable reference for the 0.5 mm offset. It is clear from the Results that the wall thickness can have a major effect on the plaque stress. Further justifications are therefore needed for this assumption, otherwise the present study is just another engineering exercise without real meaning.

3. On page 7: “As the applied FE software Ansys does not offer this material option, the model was implemented via a user material subroutine”. Since this is a user-defined function, it is the authors’ responsibility to thoroughly test and validation its accuracy. Please provide evidences of both validation and verification of this user-defined function as an appendix to the manuscript.

4. On page 10: “To compare models of more patients, a completely different methodology and a statistically significant number of patients would be necessary which is out of scope of this paper.” This is a major concern to the material presented. To provide confidence to the reader, the authors may at least include one or more patient results in the analysis. By varying the input parameters for 1 patient, this study results could be bias.

5. On page 11: “In the DoE methodology, a 2-factorial design was performed and therefore axial points were included (face-centres).” Please explain what axial points (face-centres) is in layman’s terms for wider audience.

6. On page 11: “The first principal stresses were adopted as stress response indicators, especially PPS and PCS; for all the 3D models analysed, they were evaluated in cross sections with a 0.2 mm span”

a. Which is the first principal stress?

b. Why evaluated in cross-sections with a 0.2 mm span? E.g, why not 0.1 mm or 0.3 mm? Please elaborate.

7. On page 12: “Therefore, ex vivo MRI was used for the 3D reconstruction in this study” Not sure the benefit of ex vivo MRI here. It seems like a random variable to increase the complexity of the study with no relevant clinical input.

8. On page 13: “Results near the boundary conditions and the bifurcation are not considered since they cause non-realistic artefacts (stress concentrations due to boundary conditions)”. While artefects near the boundary conditions are acceptable, the effect of bifurcation cannot be ignored. Specifically, plaques are often developed at the bifurcation. The authors might want to explain this in a bit more details.

9. On page 14: Figures 7 and 8 are very difficult to interpret at first sight. Consider redrawing the figures or providing clearer captions.

10. On page 16, Table 3: It is not clear whether there is any clinical meaning to the statistics provided here given it is a single study with only 1 patient. Further, this should be in the Results section.

6. PLOS authors have the option to publish the peer review history of their article (what does this mean?). If published, this will include your full peer review and any attached files.

Reviewer #1: No

Reviewer #2: No

---

## [Author Response · Author response to Decision Letter 0]

20 Jul 2020

Journal requirements

All files were corrected to meet a PLOS ONE style requirements. All data will be available when accepted. Since we added another patient into the analysis it could not be added as supplementary data. Proper captions were included for supplementary data in our manuscript.

We would like to thank all Reviewers for their comments and suggestions. As a consequence of this review, we have added another patient and edited the text of the manuscript to strengthen and clear up the message of the study. We have also corrected the text where errors were pointed out. Suggestions have been incorporated also in the appropriate paragraphs of the ‘Introduction’, ‘Material and Methods’, ‘Results’, ‘Discussion’ and ‘Conclusion‘ sections as described below in detail. We hope to have addressed their concerns sufficiently.

Reviewer #1:

The authors have performed a finite element analysis of a carotid atherosclerotic plaque using an endarterectomy specimen.

They have assessed the effect of wall thickness, MR imaged slice thickness, lipid and fibrous tissue mechanical properties and media anisotropy. They have found that wall thickness and the MR slice thickness are not so critical as the mechanical properties of the lipid and fibrous tissues when computing the peak stress that might be responsible for plaque rupture.

Though it is an interesting exercise, a discussion on plaque characteristics found to be associated with plaque rupture and outcome in current literature is missing and would add critical information and perspective on the impact of their results. It is well known that a thin fibrous cap, a large lipid pool and intraplaque hemorrhage are marks of vulnerable plaque, but this markers are not included, or discussed, here.

We fully agree with the reviewer that discussion of our results to plaque vulnerability should be included. Therefore, we discuss now also the vulnerability criteria mentioned by the reviewer and their applicability “in vivo” and compare our results and plaque composition with some additional clinical and mechanical reviews related to this problem. Moreover, we have added another patient and the resulting stresses exceeded the globally accepted threshold value of plaque tissue strength, thus suggesting that although the strict plaque vulnerability definition is not met, high stresses may occur and cause a plaque rupture.

Other comments:

Introduction

It is not clear what the author mean with the statement "While 2D histology sections or images of autopsy samples enable us ..... or with samples from endarterectomy where the plaque is resected without media and adventitia". The endarterectomy segments would be analyzed by 2D histology, or where do the 2D histology sections come from? Do the authors need to replace "or" with "of" when they say "While 2D histology sections or images"?

The sentence was reformulated to avoid misunderstanding. Page 4, line 72-76.

The introduction should include a statement indicating how their methodology is going to overcome the listed problems found in current literature.

We accept the reviewer´s proposal and a corresponding sentence was added in the introduction section. Page 5, line 82-84.

Methods

It is unclear whether the authors scanned the endarterectomy specimen prior to histological sectioning, or before, please clarify. Also, they could comment why they did not scan the subject prior to surgery, that would have enable them to have a proper outer wall rendering.

At the beginning of „Histology“ section it is stated that histology was performed after MRI. Although a pre-operative imaging like MRI might provide information about outer wall boundaries, the boundary is hard to distinguish from other soft tissues in standard MRI scans and it is not possible to distinguish the inner wall surface when a plaque is present. Moreover, the combination of pre and post-operative imaging would require e.g. multimodal image registration as Moerman et al. [1] proposed recently with adjustments for unloaded geometry, which is out of range of the proposed study taking the scanned geometry as load-free. However, since such a combination was not ever used for computational modelling it brings potential for further studies.

The paragraph of the Methods section discussing the use of only one case and how the results can, or cannot, be transferred to other cases, should be moved to the limitations paragraph in the Discussion.

The reviewer´s comment was accepted and this section was moved into limitations. Moreover, we added another patient into this study where all the main results were confirmed.

Discussion

The authors are encouraged to add some discussion on markers for vulnerable plaque, such as the size of the lipid pool, fibrous cap thickness and how their results would be if those were considered, or on how their methodology could accommodate for those markers.

As mentioned above in the overall reviewer´s comment response, we discussed this issue within the discussion section with some additional clinical and mechanical reviews and papers. We add the reference Burke et al. [2] defining the critical fibrous cap thickness as lower than 65 µm which is below the resolution of MRI images and thus not applicable in clinics. In contrast, our methodology is seeking for parameters potentially detectable in vivo and trying to reduce the impact of the parameters which cannot be evaluated in vivo and thus create an additional criterion completing the actually applied markers. Future improvements of imaging methods or material data could improve quality of our models and their predictions more easily then to reach an in vivo detectability of fibrous cap thickness close to the critical value. The lipid pool size was considered in advance in the selection of suitable patients as mentioned in the methodology section. Page 17-18, line 341-352.

In addition, the authors are encouraged to revise their use of English, there are some grammar and spelling errors that, at times, lead to statements that are incorrect, such as "... lead to the stroke in the carotid arteries", when stroke might be due to carotid stenosis but does not occur in those vessels, but in the brain.

We accepted the reviewer´s proposal and a thorough English revision was performed.

Reviewer #2:

Authors Lisický et al. present a manuscript titled “Consideration of stiffness of wall layers is decisive for patient-specific analysis of carotid artery with atheroma” for possible publication in the PLOS ONE. The authors have employed a Design of Experiment concept with ex vivo imaging data to investigate the effects of various parameters on plaque stress. The authors have also performed a comprehensive analysis on a number of phantom models (50 to be exact) to potentially draw a statistically meaning conclusion. In brief, the authors have highlighted that wall stiffness are more important than wall thickness in predicting plaque stress. Although the study is fundamental sound, the manuscript suffers a few shortfalls. In particular, while the authors have quantified the effects of parameters such as wall thickness, fibrous tissue stiffness, etc., the lack of patient number means that the study is, at best, hypothesis generating. It is unclear how the results would differ even with images from one additional patient. Given the highly doubt reproducibility of the present method clinically, the manuscript cannot be recommended for publication in the PLOS ONE in its current form.

Nevertheless, I have provided a list of suggestions/comments for the authors:

We fully agree with the reviewer that our previous use of a single patient was questionable and therefore we added another patient into the analysis to strengthen our results. All the pointed comments are addressed below in detail.

1. In the Abstract: “Except for anisotropy, all the other investigated factors are taken as continuous in the range based on published experimental results”. This sentence is difficult to understand. Consider rephrase.

The sentence was reformulated to improve understanding. Page 2, line 28

2. On page 6: “Since the individual vessel wall layers cannot be distinguished in the in vivo MRI scans and the sample harvested at endarterectomy cannot include the media and adventitia layers of the wall, the outer surface of the obtained geometry was used to create the wall volume… equidistantly offset by 0.5 mm… the assumption of constant wall thickness (WT) was adopted here as an appropriate simplification and the thickness was included into DoE as one of the investigated parameters”. While the authors have provided suitable reference for the 0.5 mm offset. It is clear from the Results that the wall thickness can have a major effect on the plaque stress. Further justifications are therefore needed for this assumption, otherwise the present study is just another engineering exercise without real meaning.

Full 3D models with patient-specific geometry are taken as highest standard of computational modelling. However, in contrast to models with simplified geometry they mostly do not consider the vessel wall thickness with its specific material properties. Local variable wall thickness cannot be directly obtained from in vivo imaging resulting in some constant assumptions as mentioned in references. Our approach, assuming constant wall thickness has shown large differences in wall deformation which were not considered in previous studies. Some models without consideration of the wall gave deformations hardly believable in a real artery with atheroma (some 2 mm in internal CA) while deformation of the model with specific properties of arterial wall considered was much lower in all the investigated cases. Therefore, we believe that our assumption, although not rigorously supported, improves significantly the level of computational model and may be useful for future modelling of plaque vulnerability. Investigation of the influence of specific wall thickness might be inspiration for future works though it should be supported with experimental evaluation of wall thickness.

3. On page 7: “As the applied FE software Ansys does not offer this material option, the model was implemented via a user material subroutine”. Since this is a user-defined function, it is the authors’ responsibility to thoroughly test and validation its accuracy. Please provide evidences of both validation and verification of this user-defined function as an appendix to the manuscript.

We accepted the reviewer´s suggestion and added validation and verification via appendix (S1 appendix). For this purpose, we used experiments and predictions from a frequently cited paper dealing with coronary arteries and compared our implemented model with experimental data and original analytical model. We simulated the uniaxial tension test as the authors did in the paper and our user-defined subroutine gave a nearly identical response; the differences are far below common inaccuracies of numerical methods.

4. On page 10: “To compare models of more patients, a completely different methodology and a statistically significant number of patients would be necessary which is out of scope of this paper.” This is a major concern to the material presented. To provide confidence to the reader, the authors may at least include one or more patient results in the analysis. By varying the input parameters for 1 patient, this study results could be bias.

We accepted the reviewer´s suggestion and added another patient in the analysis. Thus the cited sentence was omitted and the corresponding parts of the text were changed and highlighted in the text. Main conclusions based on the first patients were paralleled by the additional patient. The previous results were really partially biased but only in the case of MRI slice thickness factor which was found significant for sample 2. Here the explanation is quite straightforward, a specific geometry of the plaque and its components may play a key role in plaques and locations where a locally thin fibrous cap occurs and may not be included when a high slice thickness is concerned. All the other conclusions were confirmed also with the additional patient.

5. On page 11: “In the DoE methodology, a 2-factorial design was performed and therefore axial points were included (face-centres).” Please explain what axial points (face-centres) is in layman’s terms for wider audience.

The sentence was reformulated, and the methodology was explained in greater detail. Page 10-11, line 204-207.

6. On page 11: “The first principal stresses were adopted as stress response indicators, especially PPS and PCS; for all the 3D models analysed, they were evaluated in cross sections with a 0.2 mm span”

a. Which is the first principal stress?

First principal stress is a term quite common in mechanics and it expresses the maximal (tensile) stress which occurs in any direction in the investigated point or even whole computational model. For non-technical readers, the term

maximal was added in parentheses. Page 11, line 209.

b. Why evaluated in cross-sections with a 0.2 mm span? E.g, why not 0.1 mm or 0.3 mm? Please elaborate.

The choice 0.2 mm span relates to the axial resolution of the MRI images, because within this distance all values (geometrical as well as mechanical) are numerically interpolated, thus no extreme can occur in between. In our best model axial resolution was 0.25 mm, and finite element size is 0.3 mm thus this span would be also sufficient to capture any maxima.

7. On page 12: “Therefore, ex vivo MRI was used for the 3D reconstruction in this study” Not sure the benefit of ex vivo MRI here. It seems like a random variable to increase the complexity of the study with no relevant clinical input.

We are not sure here whether the reviewer means benefit against histology or against in vivo MRI. Evidently, ex vivo MRI is fast and enables us to obtain the patient-specific plaque geometry distinguishing individual components. In contrast, creation of histological slices is very lab-time consuming and suffers from shape distortions during the histological processing, thus it is less suitable than ex vivo MRI. The benefit against standard in vivo MRI is in better axial and in-plane resolution. The choice of the slice thickness as one of the investigated parameters was motivated by the need to test, whether a lower axial resolution gives correct results. As the in-plane resolution of in vivo MRI scanning can be improved e.g. with using a special carotid coil, as mentioned in the manuscript, information on the needed axial resolution is important just for clinical application. A high in vivo MRI axial resolution may be inappropriate when reconstructing plaques in a length of ca 20 mm. That was the reason for slice thickness factor inclusion, which can be varied easily for endarterectomy samples. Since an additional patient showed that a slice thickness may play an important role when investigating stresses, we think that focus on this problem should be even extended in future work. Variations in slice thickness in vivo cause longer scanning time which was found unsuitable for patients and made it impossible to have representative images. Even though, studies [3,4] found in vivo MRI suitable to distinguish plaque components, they also mentioned that some images were inappropriate and were excluded from evaluation. That would not allow reconstructing geometry model.

8. On page 13: “Results near the boundary conditions and the bifurcation are not considered since they cause non-realistic artefacts (stress concentrations due to boundary conditions)”. While artefects near the boundary conditions are acceptable, the effect of bifurcation cannot be ignored. Specifically, plaques are often developed at the bifurcation. The authors might want to explain this in a bit more details.

In both evaluated patients, no maximal stresses were found near the bifurcation. Complex geometry as carotid apex is, may cause problems in discretization (mostly in contact with vessel wall) and non-realistic concentrations may appear. However, with element size of 0.3 mm used in our study the critical region would not exceed this value. In case where stress concentration occurs finite element mesh is refined to ensure proper global results.

9. On page 14: Figures 7 and 8 are very difficult to interpret at first sight. Consider redrawing the figures or providing clearer captions.

We accepted the reviewer´s suggestion and the captions were reformulated. Moreover, images were slightly amended to make the representation clearer. Page 15 and 17.

10. On page 16, Table 3: It is not clear whether there is any clinical meaning to the statistics provided here given it is a single study with only 1 patient. Further, this should be in the Results section.

We accepted the reviewer´s suggestions and extended the study by one more patient. However, the proposed statistics is related to a specific patient separately as it results from the used methodology. The second patient showed similar results, thus pointing out certain regularities which should not be overlooked.

We thank you again for your consideration of our revised manuscript.

1. Moerman AM, Dilba K, Korteland S, Poot DHJ, Klein S, van der Lugt A, et al. An MRI-based method to register patient-specific wall shear stress data to histology. PLoS One. 2019;14: e0217271.

2. Burke F. CORONARY RISK FACTORS AND PLAQUE MORPHOLOGY IN MEN WITH CORONARY DISEASE WHO DIED SUDDENLY. New Engl J Med Coron. 1997.

3. Saam T, Ferguson MS, Yarnykh VL, Takaya N, Xu D, Polissar NL, et al. Quantitative evaluation of carotid plaque composition by in vivo MRI. Arterioscler Thromb Vasc Biol. 2005;25: 234–239. doi:10.1161/01.ATV.0000149867.61851.31

4. Hatsukami TS, Ross R, Polissar NL, Yuan C. Visualization of fibrous cap thickness and rupture in human atherosclerotic carotid plaque in vivo with high-resolution magnetic resonance imaging. Circulation. 2000;102: 959–964. doi:10.1161/01.CIR.102.9.959

---

## [Decision Letter · Decision Letter 1]

19 Aug 2020

PONE-D-20-04059R1

Consideration of stiffness of wall layers is decisive for patient-specific analysis of carotid artery with atheroma

PLOS ONE

Dear Dr. Lisický,

Thank you for submitting your manuscript to PLOS ONE. After careful consideration, we feel that it has merit but does not fully meet PLOS ONE’s publication criteria as it currently stands. Therefore, we invite you to submit a revised version of the manuscript that addresses the points raised during the review process.

We look forward to receiving your revised manuscript.

Kind regards,

Fang-Bao Tian

Academic Editor

PLOS ONE

Reviewers' comments:

Reviewer's Responses to Questions

**Comments to the Author**

1. If the authors have adequately addressed your comments raised in a previous round of review and you feel that this manuscript is now acceptable for publication, you may indicate that here to bypass the “Comments to the Author” section, enter your conflict of interest statement in the “Confidential to Editor” section, and submit your "Accept" recommendation.

Reviewer #1: All comments have been addressed

Reviewer #2: (No Response)

2. Is the manuscript technically sound, and do the data support the conclusions?

Reviewer #1: Yes

Reviewer #2: Yes

3. Has the statistical analysis been performed appropriately and rigorously? 

Reviewer #1: I Don't Know

Reviewer #2: No

4. Have the authors made all data underlying the findings in their manuscript fully available?

Reviewer #1: Yes

Reviewer #2: Yes

5. Is the manuscript presented in an intelligible fashion and written in standard English?

Reviewer #1: Yes

Reviewer #2: No

6. Review Comments to the Author

Reviewer #1: (No Response)

Reviewer #2: The authors have addressed most of my suggestions and the quality of the manuscript has improved. There are still a few minor concerns and I would truly appreciate if the authors can briefly address these issues:

1) A statement or two explaining the limitation of using ex vivo data in the present analysis. In particular, briefly explain how ex vivo data may be different from in vivo data once the artery/tissue leaves the human body.

2) While it is assuring that the second patient's results are within the predicted ballpark, I remain concern the term "statistical significant" in the analysis.

I suggest refining this statement. "Despite this limitation, comparable statistical significance of all results was achieved except for MRI slice thickness factor which is strongly related to the complexity of geometry and its inter-patient variability."

3) The authors may want to carefully proof read the manuscript. Here is a non exclusive list of typos/grammatical errors: line 83; lines 154-155; line 263: define ANOVA; lines 295-296; line 362 (58 divided by 147 kPa?); line 422; line 468, etc.

7. PLOS authors have the option to publish the peer review history of their article (what does this mean?). If published, this will include your full peer review and any attached files.

Reviewer #1: No

Reviewer #2: No

---

## [Author Response · Author response to Decision Letter 1]

4 Sep 2020

Thank you very much for the comments on our manuscript entitled “Consideration of stiffness of wall layers is decisive for patient-specific analysis of carotid artery with atheroma” (PONE-D-20-04059R1). We have addressed the remaining minor concerns of Reviewer 2 as follows:

The authors have addressed most of my suggestions and the quality of the manuscript has improved. There are still a few minor concerns and I would truly appreciate if the authors can briefly address these issues:

1) A statement or two explaining the limitation of using ex vivo data in the present analysis. In particular, briefly explain how ex vivo data may be different from in vivo data once the artery/tissue leaves the human body.

As required, explanation of differences between geometries based on in vivo and ex vivo imaging is added in Limitation Section where the consequences are discussed in greater detail now. Page 23, line 471-483.

2) While it is assuring that the second patient's results are within the predicted ballpark, I remain concern the term "statistical significant" in the analysis.

I suggest refining this statement. "Despite this limitation, comparable statistical significance of all results was achieved except for MRI slice thickness factor which is strongly related to the complexity of geometry and its inter-patient variability."

The statement was reformulated to reflect better the limitations given by the low number of patient-specific geometries. Page 23, line 484-489.

3) The authors may want to carefully proof read the manuscript. Here is a non exclusive list of typos/grammatical errors: line 83; lines 154-155; line 263: define ANOVA; lines 295-296; line 362 (58 divided by 147 kPa?); line 422; line 468, etc.

Careful proofreading of the manuscript was done and all the mentioned (and some other) typos were corrected.

---

## [Editor Report · Decision Letter 2]

7 Sep 2020

Consideration of stiffness of wall layers is decisive for patient-specific analysis of carotid artery with atheroma

PONE-D-20-04059R2

Dear Dr. Lisický,

We’re pleased to inform you that your manuscript has been judged scientifically suitable for publication and will be formally accepted for publication once it meets all outstanding technical requirements.

Kind regards,

Fang-Bao Tian

Academic Editor

PLOS ONE
---

## [Editor Report · Acceptance letter]

15 Sep 2020

PONE-D-20-04059R2

Consideration of stiffness of wall layers is decisive for patient-specific analysis of carotid artery with atheroma

Dear Dr. Lisický:

I'm pleased to inform you that your manuscript has been deemed suitable for publication in PLOS ONE. Congratulations! Your manuscript is now with our production department.

Kind regards,

on behalf of

Dr. Fang-Bao Tian 

Academic Editor

PLOS ONE